# Intentions of Patients with Hypertension to Receive a Booster Dose of the COVID-19 Vaccine: A Cross-Sectional Survey in Taizhou, China

**DOI:** 10.3390/vaccines10101635

**Published:** 2022-09-29

**Authors:** Chen-Qian Ying, Xiao-Qing Lin, Li Lv, Yan Chen, Jian-Jun Jiang, Yun Zhang, Tao-Hsin Tung, Jian-Sheng Zhu

**Affiliations:** 1Department of Infectious Diseases, Taizhou Hospital of Zhejiang Province, Wenzhou Medical University, Linhai 317000, China; 2Department of Infectious Diseases, Taizhou Hospital of Zhejiang Province, Zhejiang University, Linhai 317000, China; 3Department of Cardiovascular Diseases, Taizhou Hospital of Zhejiang Province, Zhejiang University, Linhai 317000, China; 4Evidence-Based Medicine Center, Taizhou Hospital of Zhejiang Province, Wenzhou Medical University, Linhai 317000, China

**Keywords:** hypertension, COVID-19, willingness-to-vaccine, China

## Abstract

COVID-19 patients with hypertension have increased hospital complications and mortality rates. Moreover, these patients also have lower antibody titers after receiving the coronavirus disease (COVID-19) vaccine. Therefore, patients with hypertension should receive a COVID-19 vaccine booster. To promote the uptake of COVID-19 vaccine booster among hypertensive patients, this study investigated patients’ willingness and factors that influence patients with hypertension to receive the COVID-19 vaccine booster. From July 2021 to August, 410 patients with hypertension were surveyed. Overall, 76.8% of patients were willing to receive the COVID-19 vaccine booster, as 82.7% of patients without comorbidities and 72.7% of patients with comorbidities were willing to receive the vaccine booster. The main factors that influenced the willingness of patients with hypertension to receive a booster dose were the preventive effect of the vaccine (χ^2^ = 52.827, *p* < 0.05), vaccine safety (χ^2^ = 42.423, *p* < 0.05), vaccine knowledge (χ^2^ = 7.831, *p* < 0.05), presence of comorbidities (χ^2^ = 4.862, *p* < 0.05), disease control (χ^2^ = 5.039, *p* < 0.05), and antihypertensive treatments (χ^2^ = 12.565, *p* < 0.05). This study’s findings highlight the need to promote knowledge about booster vaccination among patients and health management. These measures would improve patients’ willingness and knowledge about the vaccine and their health status, which are the main factors that influence patients’ intention to receive booster vaccines.

## 1. Introduction

Since the end of 2019, COVID-19 has been rampant, as of September 2022, over 600 million people have been infected and over 6 million have died (COVID-19 is a disease caused by a new coronavirus called SARS-CoV-2. The World Health Organization first learned of this new virus on 31 December 2019, following a report of a cluster of cases of “viral pneumonia” in Wuhan, People’s Republic of China [1]). The COVID-19 vaccine plays an important role in limiting the virus circulation [2], and studies have shown that antibody titers induced by COVID-19 vaccines decrease over time, and the decrease in antibody titers leads to an increase in the risk of disease [3]. In addition, study have shown that the effectiveness of the COVID-19 vaccine decreased after six months of vaccination, so the incidence of COVID-19 is lower among those who receive the booster vaccine [4]. Currently, the booster doses of COVID-19 vaccine used in China have mainly inactivated COVID-19 vaccine (Vero Cell) and adenovirus vector vaccine (Ad5-nCoV); according to the needs of national epidemic prevention and control, the booster dose can be given 6 months after the full vaccination of the above-mentioned vaccine [5]. With the gradual development and clinical application of booster vaccines and the research on chronic diseases [6], research on booster vaccines for chronic diseases is also in progress, including in patients with chronic liver disease [7], renal transplant [8], cancer [9,10], and SLE (systemic lupus erythematosus) [11]. These studies have shown that COVID-19 booster shots can increase antibody titers and reduce the rate of new coronavirus infections in these patients. By September 2022, about 89% of people had received all doses prescribed by the initial vaccination protocol in China, while only about 56% of people had received booster shots; moreover, around the world, about 62% of people had received full doses, while only about 30% people had received booster shots or only 49% of the former [12]. It shows that increasing people’s willingness to be vaccinated is crucial. It has also been reported that most of patients infected with COVID-19 have hypertension [13,14,15] and these patients have increased mortality and hospital complications [16,17,18,19]. Moreover, compared to non-hypertensive individuals, patients with hypertension have lower antibody titers to NeoCon after vaccination with the NeoCon vaccine [20]. Booster vaccination is important in patients with hypertension. However, there are limited studies on the willingness of patients with hypertension to receive a booster dose of the new crown vaccine, so we conducted a cross-sectional survey of these patients. To promote the prevalence of vaccine boosters in the hypertensive population and then control the spread of COVID-19, this study aimed to investigate the willingness of patients with hypertension to receive the vaccine booster and to identify those factors that influence these patients to receive booster vaccines.

## 2. Methods

### 2.1. Research Design and Data Collection

We conducted an anonymous, cross-sectional, population-based online survey using China’s largest online survey platform, Questionnaire Star (Ranxing Information Technology Co., Ltd., Changsha, China). Our target population was people with hypertension living in Taizhou who received a survey invitation via WeChat. According to the data of the seventh national census in 2020, the total population of Taizhou was 6,622,888 (based on the standard time point of midnight on 1 November 2020). [21] At the end of September 2021, Taizhou had more than 700,000 patients with hypertension [22]. The participants received two complete doses of inactivated COVID-19 vaccine (Vero Cell) at at least 6 months, and were able to receive a booster shot of the COVID-19 vaccine. Respondents volunteered to answer the questionnaire by scanning QRs code on their smartphones between 10 July and 9 August 2021. The inclusion criteria for this study were patients with hypertension who received an invitation and voluntarily completed the questionnaire. A total of 557 patients participated in our study in the Department of Cardiovascular Medicine, Taizhou Hospital (Taizhou, China), the largest hospital at Taizhou City, and 410 patients submitted complete and equal answers, with a response rate of 73.6%. This study was approved by the Ethics Committee of Taizhou Hospital, Zhejiang Province, China (approval number: K20210705). All the procedures were conducted in accordance with the guidelines of our institutional ethics committee and adhered to the principles of the Declaration of Helsinki. The information on all participants was kept anonymous.

### 2.2. Structured Questionnaires and Measurements

An online self-administered questionnaire was designed. The questionnaire began by explaining the background and purpose of the survey and stating that the questionnaire was answered anonymously and voluntarily with informed consent. The contents of the questionnaire consisted of five parts: (1) basic demographic information and allergic reaction, (2) information related to hypertension, (3) risk perception of COVID-19, (4) knowledge of the COVID-19 vaccine, (5) willingness to receive a booster dose of the COVID-19 vaccine. Analysis: Almost all questions were closed-ended and provided checkboxes for the responses.

Basic demographic information and allergic reaction: Sociodemographic factors included sex, age group (18–59, 60–93 years), area of residence (rural, urban), education level (junior high school and below, high school and above), occupation (farmer, worker, other). Participants were also asked whether they had allergic reaction after they received a vaccine (any vaccine). 

Information related to hypertension: Overall knowledge about the information related to hypertension was measured using the following questions: “How is your blood pressure control? (good control and stable blood pressure, poor control and unstable blood pressure)”; “Do you currently have any treatment? (yes, no)”; “Do you have diabetes, hyperlipidemia, hyperuricemia etc., (yes, no)”.

Risk perception of COCID-19: The question regarding the perceived risk of COVID-19 was: “How much do you think you are at risk of being infected with COVID-19?” Five options were provided for responses: very high, high, medium, low, or very low. In the final analysis, the first two options were categorized as “high” and the last two as “low”. 

Knowledge of the COVID-19 vaccine: Information related to the knowledge of the COVID-19 vaccine was collected through several questions: “How much do you know about the COVID-19 vaccine booster?” Five options were provided for responses: very well, well, relatively moderate, unknown, or not at all. “How safe do you think the current COVID-19 vaccines are?” The options were based on a 5-point Likert scale [23]: very safe, safe, moderate, unsafe or very unsafe. “How effective do you think the vaccine is in preventing COVID-19 infection”. The options were based on a 4-point Likert scale [23]: great, relatively great, moderate, or little. For simplification, the responses to the knowledge, safety, and preventive effect of the COVID-19 vaccine were recorded during the final analysis, with the first two options recoded as high (useful), and the others recoded as low (useless). In addition, the interviewees were asked, “When you are not sure whether you can get the COVID-19 vaccine, have you actively consulted others?” The response options were yes and no.

Willingness to receive a booster dose of the COVID-19 vaccine: Willingness to receive the booster dose of the COVID-19 vaccine was measured using the following question, “Would you like to receive a COVID-19 booster shot to prevent the virus?” Five options were provided for responses: very willing, willing, unwilling, and very unwilling. The responses were remerged during the final analysis, the first two options were merged into “willing”, and the others were merged into “unwilling”.

### 2.3. Statistical Analysis

The primary outcome of the survey was the willingness of patients with hypertension to receive a booster dose of the COVID-19 vaccine. The counts and frequency distributions of categorical data are shown, and differences between reluctance and willingness to be vaccinated were compared using the χ^2^ (chi-square) test. The dependent variable “Are you willing to get the future vaccination?” had two possible values: 0 = (“willing to be vaccinated”); 1 = (“unwilling to be vaccinated”). A χ^2^ (chi-square) test was used to assess potential factors associated with willingness to receive the COVID-19 vaccine booster in patients with hypertension, such as sex, age, place of residence, education, presence of comorbidities, disease stability, and knowledge of and attitudes toward the COVID-19 vaccine. Binary logistic regression was then performed to identify those factors that were associated with the willingness to receive the COVID-19 vaccine booster among patients with hypertension, with a calculated dominance ratio (OR) and 95% confidence interval (CI). In univariate analysis, variables significant at a level of *p* < 0.05 were included in the model. All data were analyzed using the IBM SPSS Statistics software (version 22.0; SPSS Inc., Chicago, IL, USA). Statistical significance was set at *p* < 0.05.

## 3. Results

In this study, data from 410 questionnaires were analyzed. The demographic characteristics of patients with hypertension are shown in Table 1; 55.4% were male, and 44.6% were female aged 18–93 years, mean age 56.43 years. Among all the participants, 65.4% resided in rural; 67.6% had junior high school education or below; 53.9% were a famer and 18.3% were a worker. Information related to hypertension was also recorded. The disease was stable in 363 (88.5%) patients, and antihypertensive treatment was administered in 345 (84.1%) patients at the time of the survey. Over half of all the patients had at least one comorbidity. Among the patients with comorbidities, 29.3% had diabetes, 30.7% had hyperlipidemia, and 34.6% had hyperuricemia. 

As shown in Table 2, the factors influencing the willingness to receive a booster dose of the COVID-19 vaccine in patients with hypertension were confidence in the effectiveness of the COVID-19 vaccines (χ^2^ = 52.827, *p* < 0.05), knowledge of COVID-19 vaccines (χ^2^ = 7.831, *p* < 0.05), confidence in safety of the COVID-19 vaccines (χ^2^ = 42.423, *p* < 0.05), comorbidities (χ^2^ = 4.862, *p* < 0.05), disease control (χ^2^ = 5.039, *p* < 0.05), and treatment (χ^2^ = 12.565, *p* < 0.05), while there were no differences between those who were willing and unwilling to receive the COVID-19 vaccine booster in terms of sex, age, place of residence, occupation, and education level (*p* > 0.05).

As shown in Figure 1, most respondents (n = 315 [76.8%]) were willing to receive a booster dose of the COVID-19 vaccine. Figure 2 shows that 82.7% of patients without comorbidities were willing to receive a booster dose and 72.7% of patients with comorbidities were willing to receive a booster dose.

The influencing factors that were observed after using the logistic regression model are shown in Table 3. The prophylactic effect of the COVID-19 vaccine (high vs. low, OR = 3.028, 95% CI: 1.596–5.743), safety (high vs. low, OR = 2.738, 95% CI: 1.485–5.050), and the presence of comorbidities (no vs. yes, OR = 7.169, 95% CI: 1.642–31.296) were found to be associated with the willingness of patients with hypertension to receive booster shots.

## 4. Discussion

Studies have shown that COVID-19 vaccines are effective in preventing the spread of the virus [24], while the immune effect of COVID-19 decreases over time [25]. In response to this phenomenon, many studies have shown that the booster dose of COVID-19 vaccine can increase autoantibodies to prevent COVID-19 [26]. In addition, some studies have shown that hypertension is associated with disease severity, morbidity, and mortality in patients with COVID-19 [27]. Mohseni Afshar Z et al. [28] suggested that NeoCon vaccination should be a priority for patients with hypertension. This concern should be further integrated into vaccine advocacy and booster interventions to improve the willingness of hypertensive patients to receive booster doses of the COVID-19 vaccines. This study was conducted on patients with high blood pressure who had received two full doses of the COVID-19 vaccine, which focused on the willingness of patients with hypertension to receive a booster dose of the COVID-19 vaccine. The results showed that the willingness of patients with hypertension to receive a booster dose of the COVID-19 vaccine was lower than the willingness of the general population to receive the vaccine (91.1%) as reported in our team’s previous study [29]. The results of the study on the willingness of patients with hypertension to receive the COVID-19 vaccine varied considerably between countries; for example, the proportion of willing patients was 78.2% in France [30], 75.4% in Bangladesh [31], and 55.4% in Spain [32].

Logistic regression analysis suggested that the confidence in the effectiveness of the COVID-19 vaccines (OR = 3.028, *p* = 0.001), the confidence in the safety of the COVID-19 vaccines (OR = 2.738, *p* = 0.001), and whether comorbidities (OR = 7.169, *p* = 0.009) were significantly associated with the willingness to receive a COVID-19 booster vaccine. It meant that the main factors that influence the willingness of patients with hypertension to receive the COVID-19 vaccine booster are their perception of the COVID-19 vaccine and their health status. First, some studies reported that the perception of patients regarding the COVID-19 vaccine influences the willingness of hypertensive patients to receive the booster shot; moreover, those who believe the vaccines are effective and safe are more willing to receive the booster shot, which is consistent with the findings reported in this study [33,34]. Therefore, increased publicity and awareness of the vaccine booster may increase patients’ willingness to receive the vaccine. Second, the health status of patients with hypertension influences their willingness to receive vaccinations. Patients with comorbidities were more reluctant to receive the booster vaccine, probably because they perceived their health status to be poor and were concerned about the adverse effects and impact of vaccination on their primary disease. A study on willingness to receive the COVID-19 vaccine in a Chinese population also showed that individuals with a low self-reported health status were more reluctant to receive the vaccine [35]. A study showed that the condition and severity of hypertension were associated with self-management behaviors [36]; self-management by patients with hypertension can reduce their blood pressure [37] and improve their health status (self-management refers to an individual’s ability to manage the symptoms, treatment, physical and psychosocial consequences, and lifestyle changes inherent in a person with a chronic disease) [38]. Therefore, we can improve the health of patients with hypertension and thus their willingness to receive COVID-19 booster injections by improving their self-management of chronic diseases [39].

This study has some limitations of methodological consideration. First, the sample size was small, and the geographical scope was slightly limited, Taizhou is only a small area of China, the results may not be representative of the entire country, and it means the result may have been biased; in other words, the findings may have limited generalizability. Second, the questionnaire was an online survey, which may have suffered from selection and message bias. For example, some people who do not know how to use smartphones were excluded. Moreover, the “Hawthorne effect” is unavoidable because participants were individuals who make conscious decisions, and those patients who were interested in the topic were more likely to volunteer. Third, because subjects tend to respond to beneficial options, social desirability bias may have been introduced. Fourth, during the questionnaire survey, although the COVID-19 outbreak was nationwide, the epidemic situation in Taizhou was well controlled. Our estimates had only been studied at one point in time, by explicit assessments, and cannot be applied to consider long-term effects. In addition, the prevalence of the COVID-19 epidemic may have an impact on vaccination intentions. Further studies are needed to examine the results not only in other regions of China, but also to better investigate the relationship between different factors and intention to receive a booster dose of the COVID-19 vaccine. Finally, this was a cross-sectional study and causal relationships could not be explored.

## 5. Conclusions

At the time of recurrent epidemics of COVID-19, it is important to increase the willingness of hypertensive patients, as a high-risk group, to receive booster vaccine shots, and this study investigated the willingness of this group to receive the COVID-19 booster vaccine. The study showed that the willingness of patients with hypertension to receive the booster vaccine was 76.8%, and the main factors that affect this willingness to receive the vaccine were their knowledge of the COVID-19 vaccine and their health status. In the future, we can improve the awareness and health management of patients with hypertension to increase their willingness to receive the booster vaccine and reduce their risk of contracting the virus.

## Figures and Tables

**Figure 1 vaccines-10-01635-f001:**
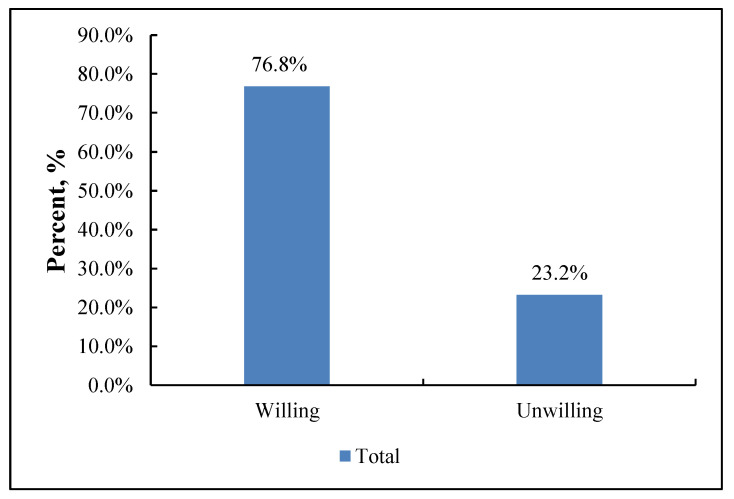
Willingness to accept the booster dose of COVID-19 vaccine, *N* = 410.

**Figure 2 vaccines-10-01635-f002:**
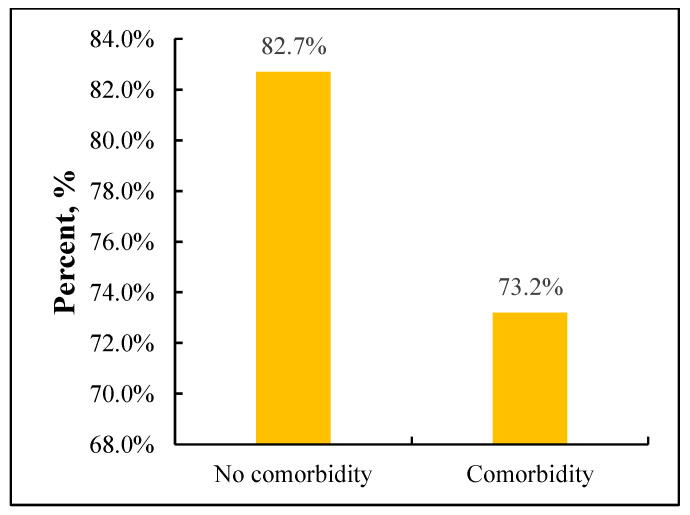
Willingness to accept the booster dose of COVID-19 vaccine, *N* = 410. *p*-value < 0.05.

**Table 1 vaccines-10-01635-t001:** Respondents’ characteristics, *N* = 410.

Independent Variables	Categories	Total Sample, N (%)	Booster COVID-19 Vaccination Acceptance			
			Willing to Be Vaccinated		Unwilling to Be Vaccinated	
		410 (100%)	315	76.8%	95	23.2%
Sex	Male	227 (55.4%)	167	73.6%	60	26.4%
	Female	183 (44.6%)	148	80.9%	35	19.1%
Age	18–59 years	235 (57.3%)	180	76.6%	55	23.4%
	60–93 years	175 (42.7%)	135	77.1%	40	22.9%
Residence	Rural	268 (65.4%)	210	78.4%	58	21.6%
	Urban	142 (34.6%)	105	73.9%	37	26.1%
Education level	Junior high school and below	277 (67.6%)	209	75.5%	68	24.5%
	High school and above	133 (32.4%)	106	79.7%	27	20.3%
Occupation	Farmer	211 (53.9%)	173	78.3%	48	21.7%
	Worker	75 (18.3%)	59	78.7%	16	21.3%
	Other	114 (27.8%)	83	72.8%	31	27.2%
Allergy history	No	360 (87.8%)	279	77.5%	81	22.5%
	Yes	50 (12.2%)	36	72.0%	14	28.0%
Diabetes	Yes	120 (29.3%)	86	71.7%	34	28.3%
	No	290 (70.7%)	229	79.0%	61	21.0%
Hyperlipidemia	Yes	126 (30.7%)	96	76.2%	30	23.8%
	No	284 (69.3%)	219	77.1%	65	22.9%
Hyperuricemia	Yes	142 (34.6%)	107	72.3%	41	27.7%
	No	268 (65.4%)	208	79.4%	54	20.6%
Comorbidity	No comorbidity	156 (38.0%)	129	82.7%	27	17.3%
	At least one comorbidity	254 (62.0%)	186	73.2%	68	26.8%
Disease control	Instability	47 (11.5%)	30	63.8%	17	36.2%
	Stability	363 (88.5%)	285	78.5%	78	21.5%
In treatment	Yes	345 (84.1%)	254	73.6%	91	26.4%
	No	65 (15.9%)	61	93.8%	4	6.2%
Risk perception of COVID-19	Very High	42 (10.2%)	37	88.1%	5	11.9%
	High	85 (20.7%)	65	76.5%	20	23.5%
	Moderate	127 (31%)	91	71.7%	36	28.3%
	Low	104 (25.4%)	83	79.8%	21	20.2%
	Very Low	52 (12.7%)	39	75.0%	13	25.0%
Proactive consultation	Yes	307 (74.9%)	243	79.2%	64	20.8%
	No	103 (25.1%)	72	69.9%	31	30.1%
Confidence in effectiveness of the COVID-19 vaccines	Useful	133 (32.4%)	115	86.5%	18	13.5%
	Possibly useful	167 (40.7%)	143	85.6%	24	14.4%
	Not sure	95 (23.2%)	53	55.8%	42	44.2%
	Useless	15 (3.7%)	4	26.7%	11	73.3%
Knowledge on the COVID-19 vaccines	Very well	80 (19.5%)	70	87.5%	10	12.5%
	Well	144 (35.1%)	114	79.2%	30	20.8%
	Relatively moderate	144 (35.1%)	106	73.6%	38	26.4%
	Unknown	39 (9.5%)	25	64.1%	14	35.9%
	Not at all	3 (0.7%)	0	0.0%	3	100.0%
Confidence in safety of the COVID-19 vaccines	Very safe	86 (21%)	76	88.4%	10	11.6%
	Safe	221 (53.9%)	184	83.3%	37	16.7%
	Moderate	86 (21%)	53	61.6%	33	38.4%
	Unsafe	16 (3.9%)	2	12.5%	14	87.5%
	Very Unsafe	1 (0.2%)	0	0.0%	1	100.0%

**Table 2 vaccines-10-01635-t002:** Univariate analysis of factors associated with booster vaccination acceptance against COVID-19, *N* = 410.

Independent Variables	Categories	Booster COVID-19 Vaccination Acceptance					
		Willing to Be Vaccinated		Unwilling to Be Vaccinated		χ^2^/t	*p*
		315	76.8%	95	23.2%		
Sex	Male	167	73.6%	60	26.4%	3.038	0.081
	Female	148	80.9%	35	19.1%		
Age	18–59 years	180	76.6%	55	23.4%	0.017	0.897
	60–93 years	135	77.1%	40	22.9%		
Residence	Rural	210	78.4%	58	21.6%	1.016	0.313
	Urban	105	73.9%	37	26.1%		
Education level	Junior high school and below	209	75.5%	68	24.5%	0.911	0.340
	High school and above	106	79.7%	27	20.3%		
Occupation	Farmer	173	78.3%	48	21.7%	1.44	0.487
	Worker	59	78.7%	16	21.3%		
	Other	83	72.8%	31	27.2%		
Allergy history	No	279	77.5%	81	22.5%	0.746	0.388
	Yes	36	72.0%	14	28.0%		
Comorbidity	No comorbidity	129	82.7%	27	17.3%	4.862	0.027
	At least one comorbidity	186	73.2%	68	26.8%		
Disease control	Instability	30	63.8%	17	36.2%	5.039	0.025
	Stability	285	78.5%	78	21.5%		
In treatment	Yes	254	73.6%	91	26.4%	12.565	0.000
	No	61	93.8%	4	6.2%		
Risk perception of COVID-19	High	102	80.3%	25	19.7%	1.256	0.262
	Low	213	75.3%	70	24.7%		
Proactive consultation	Yes	243	79.2%	64	20.8%	3.707	0.054
	No	72	69.9%	31	30.1%		
Confidence in effectiveness of the COVID-19 vaccines	High	258	86.0%	42	14.0%	52.827	0.000
	Low	57	51.8%	53	48.2%		
Knowledge on the COVID-19 vaccines	High	184	82.1%	40	17.9%	7.831	0.005
	Low	131	70.4%	55	29.6%		
Confidence in safety of the COVID-19 vaccines	High	260	84.7%	47	15.3%	42.423	0.000
	Low	55	53.4%	48	46.6%		

**Table 3 vaccines-10-01635-t003:** Factors associated with willingness to accept booster dose of COVID-19 vaccine, *N* = 410.

Independent Variables		Pa	aOR	(95% CI)	
Comorbidity	No vs. Yes	0.009	7.169	1.642	31.296
In treatment	No vs. Yes	0.112	2.464	0.811	7.490
Disease control	Instability vs. Stability	0.421	1.339	0.657	2.731
Knowledge on the COVID-19 vaccines	Low vs. High	0.245	1.412	0.789	2.528
Confidence in effectiveness of the COVID-19 vaccines	High vs. Low	0.001	3.028	1.596	5.743
Confidence in safety of the COVID-19 vaccines	High vs. Low	0.001	2.738	1.485	5.050

## Data Availability

The raw data supporting the conclusions of this article will be made available by the authors, without undue reservation.

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
