# Peer review of "Intentions of Patients with Hypertension to Receive a Booster Dose of the COVID-19 Vaccine: A Cross-Sectional Survey in Taizhou, China"

_vaccines, 2022, doi:10.3390/vaccines10101635_

Round 1

Reviewer 1 Report

I have read with interest the manuscript entitled: “Intentions of Patients with Hypertension to Receive a Booster Dose of the COVID-19 Vaccine: a Cross-sectional Survey in Taizhou, China” for the journal Vaccines, with the following comments:

1. Check the authors, two are marked with ⃰⃰ and another two with ♯

2. Abstract: It is not necessary to put background, methods, results, and conclusion. It does not indicate the type of study (cross-sectional descriptive). Rewrite it, and repeat the same sentence twice: “The willingness of patients with hypertension to receive the booster vaccination was 76.8%”

3. Introduction: correct the sentence: Since the end of 2019, COVID-19(COVID-19 is the disease caused by a new coronavirus called SARS-CoV-2. Rewrite it, it's worded incorrectly. The sentence "Studies have shown that COVID-19 incidence was lower in those who received the booster" does not mention the word vaccine. It is not explained when these patients received the first two doses of vaccines, what types of vaccine they received, and how long it had been since the second dose until the time of the booster.

4. Methods: How was the sample determined? What representativeness do you have? How many hypertensives are there in Taizhou? The sentence: "Ultimately, 410 valid data points were included in the study" are results. How many people were invited to participate? What response rate was obtained?

5. Results: The titles of the figures are written at the bottom, below them. The demographic characteristics are explained first, before the willingness to receive the booster.

6. Discussion: Repeat the phrase "The results showed that the willingness of patients with hypertension to receive a booster dose of the COVID-19 vaccine was 76.8%", this is a result. Correct the sentence: “First, the perception of the booster vaccine influenced the willingness of patients with hypertension to receive the booster, and those who believed the booster to be effective and safe were more willing to receive it, which is consistent with the results reported in this study”. The authors have correctly mentioned the limitations of their study.

Conclusion: The authors repeat again the sentence: we found that the willingness to receive the booster vaccine for hypertension was 76.8%.

References: Bibliographical references must be reviewed and written according to the journal guidelines.

The authors must present the manuscript as a brief comment and improve its structure and writing. They are recommended to rewrite it. Also,  need to improve English.

Author Response

Dear Reviewers,

Thank you for your referee comments. We thank the reviewer for useful comments and suggestions. We have substantially revised the paper as requested. The important issues raised by the reviewer have been clarified, corrected, and elaborated. Our manuscript has been advanced in a professional editing service. We hope the correction of the revised manuscript is satisfactory and met the requirement of Vaccines. Please find the revised manuscript and an outline of reply to the referees. We are happy to make any further change if required.

I have read with interest the manuscript entitled: “Intentions of Patients with Hypertension to Receive a Booster Dose of the COVID-19 Vaccine: a Cross-sectional Survey in Taizhou, China” for the journal Vaccines, with the following comments:

  1. Check the authors, two are marked with ⃰⃰ and another two with ♯

Response1: Thanks for the reviewer’s useful comments. We have checked the authors, * denotes two co-authors and # denotes two corresponding authors.

  1. Abstract: It is not necessary to put background, methods, results, and conclusion. It does not indicate the type of study (cross-sectional descriptive). Rewrite it, and repeat the same sentence twice: “The willingness of patients with hypertension to receive the booster vaccination was 76.8%”

Response 2: Thanks for the reviewer’s useful comments. We are very sorry for choosing the wrong article category. This is an article instead of an essay. And we have rewritten the abstract. We have removed the repeated sentence 'The willingness of patients with hypertension to receive the booster factor was 76.8%'. Please see page 1.

  1. Introduction: correct the sentence: Since the end of 2019, COVID-19(COVID-19 is the disease caused by a new coronavirus called SARS-CoV-2. Rewrite it, it's worded incorrectly. The sentence "Studies have shown that COVID-19 incidence was lower in those who received the booster" does not mention the word vaccine. It is not explained when these patients received the first two doses of vaccines, what types of vaccine they received, and how long it had been since the second dose until the time of the booster.

Response 3: Thanks for the reviewer’s useful comments. (1) We have revised this sentence. Please see page 2 lines 43-46;(2) We have revised this sentence and added the word vaccine. Please see page 2 lines 46-47;(3) We have explained when these patients received the first two doses of vaccines, what types of vaccine they received, and how long it had been since the second dose until the time of the booster. Please see page 2 lines 67-69.

  1. Methods: How was the sample determined? What representativeness do you have? How many hypertensives are there in Taizhou? The sentence: "Ultimately, 410 valid data points were included in the study" are results. How many people were invited to participate? What response rate was obtained?

 Response4:Thanks for the reviewer’s useful comments. We have explained these issues in the Methods section. At the end of September 2021, Taizhou had more than 700,000 patients with hypertension. A total of 557 patients participated in our study in the Department of Cardiovascular Medicine, Taizhou Hospital, the largest hospital at Taizhou City, and 410 patients submitted complete and equal answers, with a response rate of 73.6%.Please see page 2 lines 66-67 & 73-75.

  1. Results: The titles of the figures are written at the bottom, below them. The demographic characteristics are explained first, before the willingness to receive the booster.

Response 5: Thanks for the reviewer’s useful comments. We have revised the figures and explained the demographic characteristics first. Please see pages 3-6 tables 1-3 & figures 1-2.

  1. Discussion: Repeat the phrase "The results showed that the willingness of patients with hypertension to receive a booster dose of the COVID-19 vaccine was 76.8%", this is a result. Correct the sentence: “First, the perception of the booster vaccine influenced the willingness of patients with hypertension to receive the booster, and those who believed the booster to be effective and safe were more willing to receive it, which is consistent with the results reported in this study”. The authors have correctly mentioned the limitations of their study.

Response 6: Thanks for the reviewer’s useful comments. (1) We have removed the duplicate phrase from the discussion and modified this sentence. Please see page 6 lines 152-155; (2) We have revised the sentence. Please see page 6 lines 161-165.

Conclusion: The authors repeat again the sentence: we found that the willingness to receive the booster vaccine for hypertension was 76.8%.

Ans. Thanks for the reviewer’s useful comments. We have cut out the unnecessary sentences in the article.

References: Bibliographical references must be reviewed and written according to the journal guidelines.

Ans. We apologize for the mistake. We have reviewed and written bibliographic references according to the journal guidelines.

The authors must present the manuscript as a brief comment and improve its structure and writing. They are recommended to rewrite it. Also, need to improve English.

Ans. Thanks for the reviewer’s useful comments. We apologize for submitting the wrong type of article, this is an article instead of an essay. Our manuscript has been edited by a professional institution.

Reviewer 2 Report

This is a very interesting and meaningful flow survey result. I would like to recommend this manuscript for publication after minor revision:

1.     Page 1, Line 29, “via” should be Italics?

2.     Page 4, there is a figure that has not been mentioned in the manuscript. It this Figure 3? Please add the figure legend, and the “82.7%” and “73.2” in the figure should be revised as Times New Roman.

3.     Table 1, please check and correct the “COVID-19”.

4.     Although this article introduces the research on the vaccination willingness of hypertensive population, the occupation, income, education level and local flexible strategies for encouraging vaccination of the surveyed population are all important factors affecting the willingness. The authors can discuss it appropriately.

5.     The surveyed patients with some underlying diseases may also be cautious about vaccination, such as cancer patients <X. Sun, et al., Colorectal Cancer and Adjacent Normal Mucosa Differ in Apoptotic and Inflammatory Protein Expression, Engineered Regeneration 2 (2022) 279-287.>, Diabetes or cardiovascular disease patients <X. Zhang, et al. The role of astaxanthin on chronic diseases. Crystals 2021; 11: 505.>. Therefore, if possible, it is suggested that the author should be considerate in the Introduction or discussion.

Author Response

Dear Reviewers,

Thank you for your referee comments. We thank the reviewer for useful comments and suggestions. We have substantially revised the paper as requested. The important issues raised by the reviewer have been clarified, corrected, and elaborated. Our manuscript has been advanced in a professional editing service. We hope the correction of the revised manuscript is satisfactory and met the requirement of Vaccines. Please find the revised manuscript and an outline of reply to the referees. We are happy to make any further change if required.

Yours Sincerely

Jian-Sheng Zhu (Corresponding author)

 Reviewer 2 comments:

This is a very interesting and meaningful flow survey result. I would like to recommend this manuscript for publication after minor revision:

  1. Page 1, Line 29, “via” should be Italics?

Reponse 1: We apologize for the mistake. We have removed the wrong expression. Please see page 1 lines 29-30.

  1. Page 4, there is a figure that has not been mentioned in the manuscript. It this Figure 3? Please add the figure legend, and the “82.7%” and “73.2” in the figure should be revised as Times New Roman.

Response2: Thanks for the reviewer’s useful comments. (1) We have revised the figures and tables.   (2) We have revised the font. Please see page 6 figure 2.

  1. Table 1, please check and correct the “COVID-19”.

Response 3: We apologize for the mistake. We have revised the tables.

  1. Although this article introduces the research on the vaccination willingness of hypertensive population, the occupation, income, education level and local flexible strategies for encouraging vaccination of the surveyed population are all important factors affecting the willingness. The authors can discuss it appropriately.

Response 4: Thanks for the reviewer’s useful comments. Our research idea is to conduct univariate analysis first, and then regression analysis of the factors with differences. In table 2 we found that there were no differences between those who were willing and unwilling to receive the COVID-19 vaccine Booster In terms of sex, Age, place of residence, occupation, and education level (P>0.05), therefore they were not included in the regression models.

  1. The surveyed patients with some underlying diseases may also be cautious about vaccination, such as cancer patients <X. Sun, et al., Colorectal Cancer and Adjacent Normal Mucosa Differ in Apoptotic and Inflammatory Protein Expression, Engineered Regeneration 2 (2022) 279-287.>, Diabetes or cardiovascular disease patients <X. Zhang, et al. The role of astaxanthin on chronic diseases. Crystals 2021; 11: 505.>. Therefore, if possible, it is suggested that the author should be considerate in the Introduction or discussion.

Reponse 5:Thanks for the reviewer’s useful comments. We have read these two articles and found them very valuable for reference. We have cited them in the article. Please see references 3 & 7.

Round 2

Reviewer 1 Report

The authors have improved the quality of the manuscript and have incorporated the suggestions and recommendations issued so that it could be publishable.